# Masses of Hadrons, Tetraquarks, and Pentaquarks Through a Tsallis-Entropy Approach in the MIT Bag Model

**DOI:** 10.3390/e27070681

**Published:** 2025-06-26

**Authors:** Vanesa Fernández, Gerardo Herrera Corral, Manuel A. Matías Astorga, Jesús M. Sáenz

**Affiliations:** 1Departamento de Física y Matemáticas, Universidad Autónoma de Ciudad Juárez, Ciudad Juarez 32310, Mexico; al203553@alumnos.uacj.mx (V.F.); jessaenz@uacj.mx (J.M.S.); 2Departamento de Física, Center for Research and Advanced Studies, Mexico City 07360, Mexico; alejandromatiasastorga@gmail.com

**Keywords:** hadron, spectroscopy, Tsallis, entropy, bag model, QCD, pentaquarks, tetraquarks

## Abstract

We propose a new approach to describe hadrons and new forms of quark matter in the MIT bag model. The proposal is an extension that integrates a Tsallis nonextensive statistics description of quarks and gluons and is shown to capture the main features underlying mass spectroscopy. While the traditional bag model successfully accounts for the masses of light hadrons, it has not been widely used to describe exotic quark configurations. We compute the system’s internal energy and derive expressions for hadronic masses, showing how they depend on the Tsallis parameter. The model reasonably reproduces known mass spectra of conventional hadrons and naturally extends to exotic states such as tetraquarks and pentaquarks. Additionally, we provide estimates of hadron radii within this framework. Our results suggest that nonextensive statistics provide a meaningful generalization of the bag model, capturing both conventional and exotic features of hadronic matter within a unified formalism.

## 1. Introduction

There is currently intense theoretical and experimental activity focused on estimating and measuring the masses, decay widths, structures, and properties of hadrons and new forms of quark matter [1]. The renewed interest in phenomenological hadron spectroscopy follows the recent discovery of exotic hadronic states [2,3,4,5,6,7,8,9,10,11,12,13,14].

Hadron mass spectra have long been studied in a variety of theoretical frameworks. A renewed effort, driven by new observations, is now reviewing some of these ideas. Quark models aim to identify the composition of hadronic systems and describe their interactions using specific quark configurations and potentials. For instance, in [15], a system of two heavy quarks is modeled as a non-relativistic system governed by either a Coulomb-plus-linear or a logarithmic potential. The authors utilize the experimental value of the double-charm $\Xi_{cc}$ baryon mass and then extend the model by incorporating two light quarks. Similarly, in [16], the heavy quark limit is applied, treating the system as a nucleus formed by two compact heavy-color triplets with light quarks bound to a stationary color charge. A comparable approach was previously explored in reference [17], where diquonia systems, consisting of two quarks and two antiquarks, are treated as non-relativistic systems under specific potentials. In [18], a similar system is analyzed using a Coulomb potential, a confining term, spin-spin interactions, and relativistic corrections. Likewise, in [19], a constituent quark model investigates tetraquark states, calibrating parameters with masses of heavy hadrons containing the relevant quarks and employing a non-relativistic Hamiltonian for the constituent quarks. Numerous other studies follow similar approaches [20,21,22,23].

Instead of a model-dependent treatment using constituent quarks, one may use QCD Sum Rules [24] to determine the masses of hadrons. Additionally, the Chromomagnetic Interaction Model (CMI) [16,17] provides another perspective on the interactions within these systems.

Hadronic matter is also studied with Lattice QCD, which provides a nonperturbative approach to solving QCD on a discretized spacetime lattice [25]. However, it is well known that this approach covers only part of the phase diagram at zero or very low chemical potentials due to the so-called sign problem. This problem arises from the non-Hermitian nature of the Dirac operator at finite chemical potentials, which makes Monte Carlo sampling difficult. This limitation prevents the full exploration of deconfinement under extreme conditions of temperature and density.

The MIT bag model [26,27,28,29] is widely used to describe hadronic matter. In this model, hadrons are treated as extended objects composed of quarks confined within a finite volume, with confinement modeled through a constant external bag pressure. While gluons are not explicitly included as constituent particles inside individual hadrons, their collective contribution to the system’s thermodynamical properties (e.g., pressure and entropy) can be incorporated statistically. In our approach, we generalize the MIT bag model by integrating Tsallis nonextensive statistics, allowing us to introduce effective correlations between quark and gluon subsystems [30]. Such modified bag models enable the study of quark matter at high temperatures and/or baryonic densities. In this regime, hadrons undergo deconfinement, giving rise to a new phase of matter known as quark–gluon plasma [31,32], which is of particular interest in heavy-ion collisions.

The structure of this article is as follows: Section 2 introduces the theoretical foundation, including the original MIT bag model and the fundamentals of Tsallis entropy. Section 3 presents the results and discusses the mass estimates for various hadronic systems. Finally, in Section 4, we conclude and summarize the physical implications of our findings. We briefly outline future perspectives on future applications of nonextensive statistical models in hadron physics.

## 2. Phenomenological Framework

### 2.1. The MIT Bag Model

The first description of the bag model was presented by P.N. Bogoliubov [33]. However, the primary defect of this model was the violation of energy–momentum conservation. Other models were then proposed [34] in which quarks are treated as massless particles inside a bag of finite dimensions. Confinement results from the balance between the pressure exerted by quarks and the bag pressure, which was introduced as a phenomenological quantity that accounts for the nonperturbative effects of QCD. Including the external pressure resolved the issue of energy–momentum conservation violation in Bogoliubov’s pioneer work.

This MIT model enables accurate estimations of hadron properties, including charge radii, magnetic moments, and masses. Moreover, the values of critical temperature and chemical potentials at which the phase transition occurs were also estimated.

### 2.2. Tsallis Statistics

Boltzmann–Gibbs (BG) statistics work perfectly for classical systems with short-range forces and relatively simple dynamics in equilibrium. Tsallis proposed a generalization of the BG statistics [35,36,37,38]. In the new frame, entropy is not an extensive property.

Tsallis statistics have been successfully applied to complex systems. For them, the formalism of nonextensive statistical mechanics is a helpful concept that incorporates correlations between subsystems. A more general Boltzmann factor is introduced, which depends on an entropic index q and which, for *q* = 1, reduces to the ordinary Boltzmann–Gibbs statistics [35].

In conventional BG statistics, we use the entropy from information theory, i.e., $S_{BG}=-k_{B}\sum_{i=1}^{W}p_{i}lnp_{i}$, where $p_{i}$ is the probability associated with an event and W∈N is the total number of possible (microscopic) configurations. Tsallis postulates that, in nonextensive systems, the entropy is(1)$$S_{q}\equiv k_{B}\frac{1-\sum_{i=1}^{W}p_{i}^{q}}{q-1};\;(q\in R)$$
where q is any real number. From Equation (1), one can obtain the BG statistics with q=1, and we can also obtain the additive property to entropy for a system with two probabilistically independent subsystems A and B (i.e., if $p_{ij}^{A+B}=p_{i}^{A}p_{j}^{B}$) [37], such as quarks and gluons inside a hadron as(2)$$\frac{S_{q}(A+B)}{k_{B}}=\frac{S_{q}(A)}{k_{B}}+\frac{S_{q}(B)}{k_{B}}+(1-q)\frac{S_{q}(A)}{k_{B}}\frac{S_{q}(B)}{k_{B}}\;$$
where, in contrast with $S_{BG}$, which is additive, entropy $S_{q}$ is non-additive for q≠1. This non-additivity will make it extensive for various classes of systems [38].

Several types of generalized stochastic dynamics have been studied, for which Tsallis statistics can be proved rigorously. Physical applications include high-energy physics [39], 2D and 3D turbulence [40,41,42,43], the statistics of cosmic rays [44,45], and many other phenomena [36].

### 2.3. Description of a Simple Hadron Model

In the MIT bag model, quarks are treated as free particles confined inside a finite region of space. The system’s stability is maintained by an external, constant pressure applied to the bag’s surface. Confinement arises from the equilibrium between the internal pressure P exerted by the quarks and gluons and the bag pressure B. This parameter, B, is a key phenomenological quantity in the model, encapsulating the nonperturbative effects of Quantum Chromodynamics (QCD), thus making it central to understanding the dynamics of confinement. The relationship between bag pressure B and radius R is:(3)$$B^{\frac{1}{4}}={\left(\frac{2.04N}{4\pi }\right)}^{\frac{1}{4}}\frac{1}{R}\;$$
where N is the number of quarks. The bag model pressure has dimensions of energy divided by volume. In natural units where ℏ=c=1, B14 has dimensions of mass. We use natural units throughout the derivations, and we express B14 in MeV and R in femtometers (fm) and GeV^−1^.

We shall consider only bags with static, spherical boundaries and quark, antiquark, and gluon systems with no interaction, with an equal number of quarks and antiquarks. At high temperatures (where quarks and gluons have large momentum), the gluons behave like an ultra-relativistic Bose gas. The pressure arising from the gluons is detailed in [46,47]; the gluon pressure is therefore given by:(4)$$P=\frac{g_{G}8\pi }{(hc{)}^{3}}(k_{B}T{)}^{4}\frac{\pi^{4}}{90}\;$$
where T is the system’s temperature. In natural units and considering the degeneracy due to gluons $g_{G}=8\times 2=16$ (where the factor of 8 is due to generators of the SU(3) group of gluons, and the factor of 2 is due to spin orientation), we have the gluon pressure:(5)$$P_{G}=g_{G}\frac{\pi^{2}}{90}T^{4}\;.$$

On the other hand, we have the creation–annihilation processes between quarks and antiquarks, and from this we treat the system as a mixed gas. Although quarks and antiquarks undergo creation–annihilation processes, we do not consider the particles and remnants of these processes, and we obtain a relation between the number of quarks and antiquarks (their difference) and their chemical potentials, $\mu_{+}=-\mu_{-}=\mu$. Additionally, we assume that momentum is large enough compared to rest mass to let m→0. Thus, the quark pressure of the system is given by(6)$$P_{Q}=\frac{g_{Q}(k_{B}T{)}^{4}}{(\hbar c{)}^{3}}\frac{1}{3}\left[\frac{7\pi^{2}}{120}+\frac{1}{4}{\left(\frac{\mu }{k_{B}T}\right)}^{2}+\frac{1}{8\pi^{2}}{\left(\frac{\mu }{k_{B}T}\right)}^{4}\right]\;.$$

Since the quark degeneracy factor is $g_{Q}=N_{f}N_{c}N_{s}=2\times 3\times 2=12$ (where the factor of 2 is due to flavor number as we consider only up and down quarks; the factor of 3 is due to color charge, and the factor of 2 is due to spin orientation), the quark pressure becomes(7)$$P_{Q}=\frac{g_{Q}}{3}\left[\frac{7\pi^{2}}{120}+\frac{1}{4}{\left(\frac{\mu }{T}\right)}^{2}+\frac{1}{8\pi^{2}}{\left(\frac{\mu }{T}\right)}^{4}\right]T^{4}\;.$$

From Equation (3), the entropy for a hadron with quarks, antiquarks, and gluons (in natural units) is:(8)$$S_{q}\left(Q+G\right)=S_{1}\left(Q\right)+S_{1}\left(G\right)+\left(1-q\right)S_{1}\left(Q\right)S_{1}\left(G\right).$$

Equation (8) is a special case in which we consider that the quark and gluon entropy is BG entropy, but the combined quark–gluon entropy is not of the BG type; this is because we are considering that particles of the same type do not self-interact, that is, quarks and antiquarks do not self-interact between them, and gluons do not interact between them, but gluons and quarks (and antiquarks) do interact between them as a first approximation. Thus, $S_{1}(Q)$ and $S_{1}\left(G\right)$ correspond to this case of no self-interaction. We can rewrite Equation (8) as(9)$$S_{q}=S_{Q}+S_{G}+\left(1-q\right)S_{Q}S_{G},$$
where $S_{Q}$ stands for quark and antiquark BG entropy, and $S_{G}$ is the gluon BG entropy. From [48], we have that (in the ultra-relativistic fermion/boson gas regime for each), the quark and gluon entropies are, respectively(10)$$S_{Q}=g_{Q}\left[\frac{7\pi^{2}}{90}+\frac{1}{6}{\left(\frac{\mu }{T}\right)}^{2}\right]VT^{3};\;S_{G}=4g_{G}\frac{\pi^{2}}{90}VT^{3}$$
where V is the bag volume. From Equations (9) and (10), the Tsallis entropy is:(11)$$S_{q}=\left[\frac{74\pi^{2}}{45}+2{\left(\frac{\mu }{T}\right)}^{2}\right]VT^{3}+\frac{128\pi^{2}}{15}(1-q)\left[\frac{7\pi^{2}}{90}+\frac{1}{6}{\left(\frac{\mu }{T}\right)}^{2}\right]V^{2}T^{6}$$

Using the Maxwell relation ${\frac{\partial S_{q}}{\partial V}|}_{T,\mu }={\frac{\partial P_{q}}{\partial T}|}_{V,\mu }$, the Tsallis pressure is:(12)$$\;P_{q}\left(T,\mu \right)=\left[\frac{7}{4}g_{Q}+g_{G}\right]\frac{\pi^{2}}{90}T^{4}+\frac{1}{12}g_{Q}{\left(\frac{\mu }{T}\right)}^{2}T^{4}\;\;\;\;\;\;\;\;\;\;\;\;\;\;\;+\frac{8\pi^{2}}{90}g_{Q}g_{G}\left(1-q\right)\left[\frac{\pi^{2}}{90}+\frac{1}{30}{\left(\frac{\mu }{T}\right)}^{2}\right]VT^{7}+C\left(V,\mu ,q\right),$$
where $C\left(V,\mu ,q\right)$ is an integration constant. Since, for q=1, the standard total pressure must be recovered, and the integration constant is:(13)$$C\left(V,\mu ,q\right)=\frac{1}{3}g_{Q}\left[\frac{1}{8\pi^{2}}\mu^{4}\right].$$

Substituting (12) into (11), the non-additive total pressure for the quark–gluon system is:(14)$$\;\;\;\;\;\;\;\;P_{q}(T,\mu )=\left[\frac{7}{4}g_{Q}+g_{G}\right]\frac{\pi^{2}}{90}T^{4}+\frac{1}{3}g_{Q}[{\frac{1}{4}\left(\frac{\mu }{T}\right)}^{2}+{\frac{1}{8\pi^{2}}\left(\frac{\mu }{T}\right)}^{4}]T^{4}+\frac{8\pi^{2}}{90}g_{Q}g_{G}(1\;-q)\left[\frac{\pi^{2}}{90}+\frac{1}{30}{\left(\frac{\mu }{T}\right)}^{2}\right]VT^{7}.$$

By using the Maxwell relation ${\frac{\partial S_{q}}{\partial V}|}_{T,\mu }={\frac{\partial P_{q}}{\partial T}|}_{V,\mu }=-\frac{\partial^{2}F_{q}}{\partial T\partial V}$, we can obtain an expression for the system’s internal energy. From this last relation, we have two expressions:(15)$${\left(\frac{\partial F_{q}}{\partial T}\right)}_{V}=-S_{q};\;{\left(\frac{\partial F_{q}}{\partial V}\right)}_{T}=-P_{q}$$
where $F_{q}$ is the Helmholtz energy using Tsallis statistics. From (14), the Helmholtz energy is:(16)$$F_{q}(T,V,\mu )=-\left[\frac{37\pi^{2}}{90}+{\left(\frac{\mu }{T}\right)}^{2}+\frac{1}{2\pi^{2}}{\left(\frac{\mu }{T}\right)}^{4}\right]VT^{4}-\frac{128\pi^{2}}{15}(1\;\;\;\;\;\;-q)\left[\frac{\pi^{2}}{90}+\frac{1}{30}{\left(\frac{\mu }{T}\right)}^{2}\right]V^{2}T^{7}+C(\mu ,q)$$
where C(μ,q) is an integration constant that only depends on q. However, we can assume that the integration constant is zero, given the free choice of the zero-point energy.

The internal energy of the hadron is given by(17)$$U_{q}\left(T,V\right)=F_{q}\left(T,V\right)+TS_{q}\left(T,V\right)\;\;\;\;\;\;\;\;\;\;\;\;\;\;\;\;=[\frac{37\pi^{2}}{30}+{\left(\frac{\mu }{T}\right)}^{2}-{\frac{1}{2\pi^{2}}\left(\frac{\mu }{T}\right)}^{4}]VT^{4}\;\;\;\;\;\;\;\;\;\;\;\;\;\;\;\;\;\;+\frac{128\pi^{2}}{15}\left(1-q\right)[\frac{{\;\pi }^{2}}{15}+{\frac{2}{15}\left(\frac{\mu }{T}\right)}^{2}]V^{2}T^{7}$$
which matches with the energy of the quarks (as an ultra-relativistic Fermi gas) and gluons (as an ultra-relativistic Bose gas) when q=1:(18)$$\;\;\;\;{E_{Q}|}_{\mu \neq 0}=g_{Q}\left[\frac{7\pi^{2}}{120}+{\frac{1}{4}\left(\frac{c}{T}\right)}^{2}+{\frac{1}{8}\left(\frac{\mu }{T}\right)}^{4}\right]VT^{4};\;\;{E_{G}|}_{\mu \neq 0}=g_{G}\frac{\pi^{2}}{30}VT^{4}$$

### 2.4. Volume and Temperature Dependence on the Bag Radius

In [49], an expression is derived for temperature as a function of the radius of a hypothetical “bag,” representing a quark–gluon plasma system in this work. The system has specified values of T and V, with a zero-chemical potential, and the mass of the nucleons approximates the total energy. Under these assumptions, the authors propose a formula for temperature:(19)$$T=0.109r^{-3/4}.$$

In Equation (19), r is the radius from the proton’s center, measured in femtometers (fm), and T in GeV. When the radius of a proton is less than 0.6 fm, the temperature approaches 170 MeV, which is close to the critical temperature at which the hadron deconfines.

It is worth mentioning that the temperature Equation (19) is derived from experimental data on energies measured at distinct radii within the proton. However, there exists a minimum radius at which the bag can remain stable. Additionally, the bag’s volume can be expressed as a function of its radius:(20)$$V=\frac{3}{4}\pi \;r^{3}.$$

### 2.5. Hadron Mass in the MIT Bag Model

The MIT bag model provides a framework for estimating the masses and other properties of hadrons by modeling them as a spherical bag [34]. The mass formula is given by:(21)$$M(R)=\sum_{i=n,s,c,b}n_{i}{⍵}_{i}+\frac{4}{3}\pi \;R^{3}B-\frac{Z_{0}}{R}+\Delta H,$$
and(22)$${⍵}_{i}=({m_{i}}^{2}+\frac{x_{i^{2}}}{R^{2}}{)}^{1/2}.$$

We will use this expression to estimate radii in tetraquarks and pentaquarks. In (20), the first term represents the kinetic energy of all quarks within the bag of radius R, and ${⍵}_{i}$ is the frequency of the lowest mode. The second term corresponds to the volume energy. The third term accounts for the zero-point energy, and ΔH denotes the short-range interactions between quarks. Also, $n_{i}$ represents the number of quarks or antiquarks. In (21), $m_{i}$ is the mass of a quark of flavor i, i.e., the light non-strange quarks (u,d), the strange quark (s), the charm quark (c), or the bottom quark (b). The dimensionless parameter $x_{i}=x_{i}(mR)$ is related to the bag radius by a transcendental eigenequation:(23)$$\tan x_{i}=\frac{x_{i}}{1-m_{i}R-({m_{i}}^{2}R^{2}+{x_{i}}^{2}{)}^{1/2}}$$

The interaction energy ΔH consists of two additional terms: $\Delta H=B_{EB}+M_{CMI}$. $B_{EB}$ is the spin-binding energy that arises from the short-range chromoelectric interaction between quarks and antiquarks. This energy becomes significant only when both quarks involved are massive and move non-relativistically.

$M_{CMI}$ represents the chromomagnetic interaction energy arising from the perturbative gluon exchange between quarks i and j. It is defined as(24)$$M_{CMI}=-\sum_{i<j}\left(\lambda_{i}ᐧ\lambda_{j}\right)\left(\sigma_{i}ᐧ\sigma_{j}\right)C_{ij}\;,$$
where $\lambda_{i}$ are the Gell-Mann matrices, $\sigma_{i}$ are the Pauli matrices, and $C_{ij}$ is the chromomagnetic parameter given by(25)$$C_{ij}=\frac{3\;{⍺}_{s}(R)\;\underline{\mu_{i}}\;\underline{\mu_{i}}I_{ij}}{R^{3}},$$
where ${⍺}_{s}(R)$ is the running strong coupling, $\underline{\mu_{i}}$ is the reduced magnetic moment without electric charge, and $I_{ij}$ is a rational function of $x_{i}$ and $x_{j}$, given explicitly in [34,50].

### 2.6. Tsallis q Parameter Fitting Method

Hadrons were grouped according to their spin to determine the parameter q. This classification is justified because, in the MIT bag model, the chromomagnetic interaction (resulting from the perturbative exchange of gluons between quarks) depends on the flavor, spin, and color wave functions. Therefore, the fits were performed to preserve this physical dependence while respecting this structure.

The value of q was obtained using Equation (17). For each hadronic group, the experimental values of mass and radius (charge radius for light hadrons or MIT radius for exotic states) were used, and a least-squares fit was performed to determine the value of q that best reproduces the experimental masses. We use experimentally extracted radii for the light hadron masses to test our ideas against observed values. When considering radii derived from the MIT model for light hadrons as well, we found no significant changes in the overall description. The uncertainty associated with each q value was estimated from the covariance matrix of the fit. These uncertainties, as well as those associated with the experimental or MIT radius, were propagated to compute the errors in the theoretical masses, which are reported in the corresponding tables and figures.

In summary, the contributions to entropy and pressure from quarks and gluons within the bag model were calculated. These quantities were then transformed into their non-additive forms using Tsallis statistics. The Helmholtz free energy and total internal energy of the quark–gluon system were subsequently computed (cf. Equations (16) and (17)). Temperature and volume were expressed as functions of the parameters r, μ, and q. The chemical potential μ was determined for each hadronic group. Subsequently, the parameter *r* was calculated for exotic hadrons (tetraquarks and pentaquarks) by solving Equation (21) (the mass expression discussed in the previous section) for r. Finally, the corresponding values of the non-extensivity parameter q were obtained through data fitting, and the hadron masses were computed using the modified energy expression given in Equation (26).

It is important to emphasize that, when q=1, the Tsallis formalism reduces to the standard Boltzmann–Gibbs statistics, and Equation (17) recovers the usual internal energy of an ideal ultra-relativistic quark–gluon gas, equivalent to the standard MIT bag model. Deviations from q=1 reflect nonextensive statistical behaviors that may capture additional correlations, collective effects, or long-range interactions among quarks and gluons inside the hadron.

## 3. Results and Discussion

In this section, we systematically study the masses and radii of hadrons and exotic matter. We also provide the QCD phase diagram for the different configurations of quark matter.

### 3.1. Hadron Mass in the Tsallis MIT Bag Model

The hadron mass is calculated in Equation (17), which assumes that the rest mass of the hadron arises from the total energy of the quarks and gluons within it. In the Tsallis formalism, correlations between the subsystems are enclosed within the Tsallis parameter q. Consequently, to determine the mass of the hadron, we solve for q, resulting in the following relation:(26)$$M=U_{q}(r,\mu ,q),$$
where r is the hadron radius and μ represents the chemical potential. This approach demonstrates that the Tsallis parameter q depends on the hadron’s radius, chemical potential, and experimental mass.

The chemical potential can be determined by equating specific expressions (Equations (3) and (14)) and solving for μ. For pressure B, we adjusted the number of quarks based on the type of hadron.

Figure 1 shows that the maximum chemical potential at T=0 reaches approximately 432 MeV for mesons and baryons, 475 MeV for tetraquarks, and 491 MeV for pentaquarks. This “maximum” condition signifies the point at which the bag disintegrates, transforming the system into a quark–gluon plasma. A moderate condition is selected to avoid deconfinement, where the temperature and chemical potential within the bag create stable conditions.

For a temperature of 110 MeV, the chemical potential is approximately 278 MeV for light hadrons, 327 MeV for tetraquarks, and 365 MeV for pentaquarks.

The radius values are sourced from experimental data provided by the Particle Data Group (PDG) [25] and several studies [51,52,53,54,55].

#### 3.1.1. Light Hadrons

Table 1 presents the characteristics of 12 light hadrons, which we categorized into four groups based on type (mesons or baryons) and spin [25]. Using Equation (17), we fitted the parameter q to the known masses of hadrons within each group, obtaining a specific value and its uncertainty for each case. With these fitted q values, we used Equation (17) to calculate the theoretical masses based on Tsallis statistics.

Table 2 compares these theoretical masses with experimental values, highlighting their alignment. Additionally, in Figure 2, we plot these results, showing that the values of q are very close to 1 (shown in the top left corner), as expected. We also observe that the meson groups exhibit slightly higher q values.

The discrepancies between experimental and theoretical masses in Table 2, although generally below 10%, may arise from several factors. Possibly, the use of a single effective Tsallis parameter q per hadron group may introduce a simplification that averages over possible spin, flavor, and spatial configuration effects.

#### 3.1.2. Tetraquarks States

The radius values were determined by solving Equation (21) for tetraquarks, as no experimental data on tetraquark radii are available. Using the experimental mass values provided by the Particle Data Group (PDG) [25], it was possible to calculate the bag radius r, as shown in Table 3, which also includes the corresponding experimental masses and their uncertainties. The PDG reports both statistical and systematic uncertainties; therefore, the uncertainties presented here are the quadratic sum of these two types of uncertainty. Additionally, we selected only eight of the nineteen published tetraquarks, as the remaining ones lack defined angular momentum (a necessary parameter for computing the ratios) [25]. It is important to note that this is not the charge radius, as discussed for light hadrons. The bag radius represents the effective size within which the quarks are confined, whereas the charge radius is a physical property related to the charge distribution inside the hadron.

The parameters $B,\;Z_{0},{⍺}_{s}$ and quark masses $m_{n}$ and $m_{s}$ are determined based on the mass spectra of the light hadrons N, Δ,ω, and ꭥ in their ground states [34]. Additionally, the quark masses $m_{c}$ and $m_{b}$, the binding energy $B_{{QQ}^{\prime }}$ (Q,Q′=s,c,b), and the chromomagnetic interaction eigenvalues are determined in [50].

As in the previous section, we divided the tetraquarks based on their spin. This grouping was done to obtain a value of q for each group, which was then used to recalculate the mass using Equation (17). Figure 3 and Table 4 show the experimental masses of tetraquarks, their theoretical T-MIT masses, and q Tsallis parameters.

The fitted q values for tetraquarks are noticeably lower than those obtained for light hadrons. This difference reflects the more complex internal structure of multiquark systems. Tetraquarks, containing two quark–antiquark pairs, may exhibit stronger correlations between the constituents, resulting in deviations from ideal extensive statistics. The Tsallis parameter q serves as an effective measure of such internal correlations and collective dynamics, with lower q values indicating stronger nonextensive behavior compared to conventional mesons and baryons.

#### 3.1.3. Pentaquarks States

To calculate the pentaquark bag radii, we followed the same steps as for the tetraquark radii and used the parameters given in [44,50,51]. However, the Particle Data Group has not yet reported the pentaquark spin values. Therefore, we performed two different fits for quarks with composition *nnncc*, one for $IJ^{P}=\frac{1}{2},{\frac{1}{2}}^{-}$ and the other for $IJ^{P}=\frac{1}{2},{\frac{3}{2}}^{-}$. For pentaquarks with composition *nnscc*, we performed a fit only for $IJ^{P}=0,{\frac{1}{2}}^{-}$.

Figure 4 and Table 5 show the *nnncc*, $IJ^{P}=\frac{1}{2},{\frac{1}{2}}^{-}$, and *nnscc*, $IJ^{P}=0,{\frac{1}{2}}^{-}$ pentaquark states. Table 6 presents the associated Tsallis parameter values. As in the case of tetraquarks, the mass values were taken from the Particle Data Group (PDG) [25], which reports both statistical and systematic uncertainties. The total uncertainty shown corresponds to the quadratic sum of both contributions.

Figure 5 and Table 7 show the *nnncc*, $J^{P}=\frac{1}{2},{\frac{3}{2}}^{-}$ and *nnscc*, $IJ^{P}=0,{\frac{1}{2}}^{-}$ pentaquark experimental masses and their bag radii. Table 8 shows the corresponding Tsallis parameter values.

The extracted q values for pentaquarks are lower than those for tetraquarks and light hadrons. In these cases, the physical meaning of q is not straightforward within a reductionist framework. Rather, q is a statistical parameter that encases correlations within the system. Its interpretation has been the subject of ongoing discussion. Some studies suggest that q may indicate the presence of long-range interactions, while others associate it with fractal-like structures in the system. However, these interpretations remain controversial, and there is no consensus beyond the generally accepted view that q statistically characterizes correlation among the subsystems of a system.

Figure 6 shows hadron masses as a function of the q-Tsallis parameter. An overall trend is observed: lighter hadrons tend to correspond to higher q values. It is also evident that exotic states generally require lower q values for their description, suggesting that quark–gluon correlations differ significantly between mesons and baryons compared to multiquark states.

## 4. Conclusions

In this work, we have proposed a Tsallis-MIT (T-MIT) bag model that extends the MIT bag model to accommodate nonextensive thermodynamics via the Tsallis statistics. The q-parameter in the new description was determined by fitting known hadron mass states, and then we proceeded to model exotic hadronic states. Within uncertainties, the T-MIT model reproduces known mass spectra of hadrons and provides physically reasonable predictions for tetraquarks and pentaquarks. This model allows us to connect phenomenological models and statistical mechanics.

The fact that light hadrons yield q≈1 indicates that the standard Boltzmann–Gibbs statistics, and therefore the conventional MIT bag model, adequately describes these systems, with little need for nonextensive corrections. In contrast, the observed deviations from q=1 exotic states suggest the presence of nonextensive effects possibly arising from additional correlations, collective dynamics, or long-range interactions among quarks and gluons within multiquark systems. These results imply that the Tsallis parameter q can serve as a useful phenomenological indicator of internal complexity in hadronic structure.

Our data show a correlation between the hadron type and the fitted q values, with exotic states exhibiting lower and more varied q values.

Pentaquark and tetraquark masses are described using smaller q values as compared to light hadrons. It is important to clarify that the parameter q is a statistical description that contains the correlations within the hadron system. The prevailing understanding is that q reflects deviations from extensivity and statistically characterizes internal correlations among the system’s constituents.

As of now, there is no theoretical justification linking the number of quarks in a hadron to corresponding q values. Thus, the relationship between the number of quarks and the value of q is not straightforward and depends on the nature of the internal hadron dynamics and correlations.

The internal structure of tetraquarks and pentaquarks remains an area of active investigation. As such, our current results are not sufficient to draw definitive conclusions about their substructure. Further investigations involving larger datasets, including hexaquark candidates, may help elucidate the precise dependencies between q and other hadronic properties.

The chemical potential used and corresponding hadron radii are consistent with known stability conditions for confined quark systems.

Future improvements could involve introducing flavor- or spin-dependent q parameters, including higher-order corrections to the internal energy.

While the present model reasonably describes the mass spectra of both light and exotic hadrons, it does not directly compute other observables, such as decay widths or charge distributions. Treating hadrons as localized, stationary bags poses challenges for describing decay processes. One proposed solution involves using a linear superposition of infinitely many bag states, as discussed in [56]. This method has been applied to specific decay systems, although a detailed application to the present framework lies beyond the scope of this work. As for charge distributions, since the bag model Lagrangian is invariant under phase transformations and the vector current is conserved, the component of the wave function must match at the bag boundary to ensure confinement. This allows the charge density to be expressed in terms of the quark wave functions, providing a basis to compute charge distributions. Several models exist for this purpose, and extending the T-MIT approach to incorporate charge distributions remains a promising direction for future work. In addition, this extended Tsallis-based framework has already been successfully applied to other problems, such as the QCD phase diagram [48] and internal nucleon pressure contributions [30].

The T-MIT framework offers a unified treatment of light and exotic hadrons within a single formalism, underscoring its potential as a valuable tool for exploring hadron spectroscopy. The methodology provides an avenue for systematically including nonextensive effects in hadrons, which could be extended to finite-temperature and finite-density scenarios relevant to heavy-ion collisions or early-universe conditions.

## Figures and Tables

**Figure 1 entropy-27-00681-f001:**
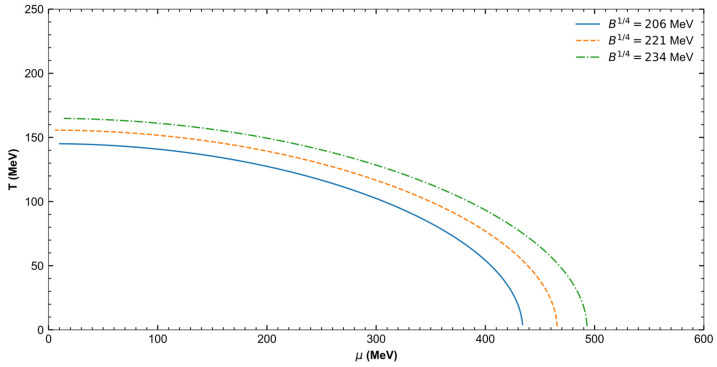
QCD phase diagram in the temperature versus chemical potential plane, for *q* = 1 and different values of bag pressure ($B^{\frac{1}{4}}$). The curves indicate the deconfinement transition for these scenarios.

**Figure 2 entropy-27-00681-f002:**
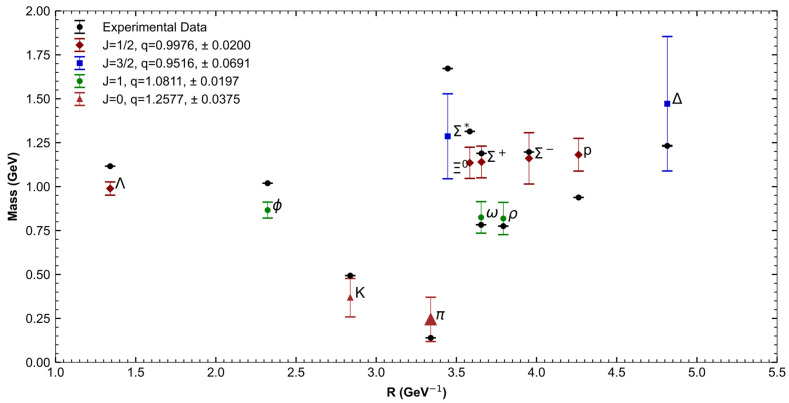
Comparison of light hadron experimental masses vs. T-MIT calculated masses and their corresponding q values.

**Figure 3 entropy-27-00681-f003:**
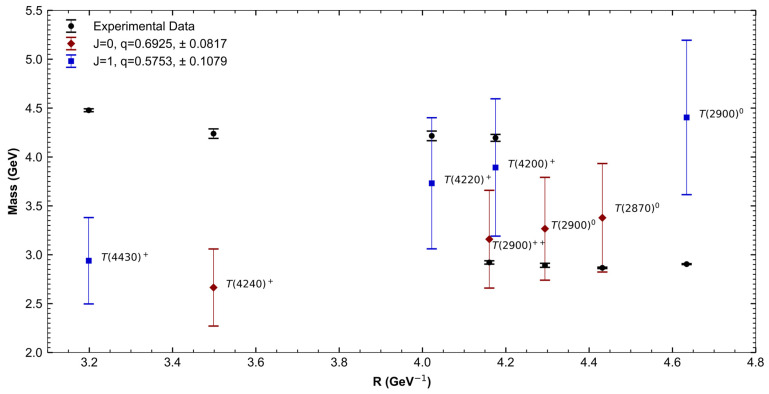
Comparison of tetraquark experimental masses vs. T-MIT calculated masses and their corresponding q values.

**Figure 4 entropy-27-00681-f004:**
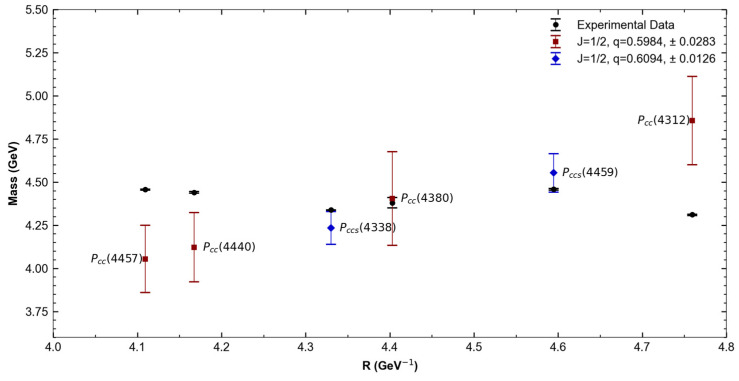
Comparison of nnncc, $IJ^{P}=\frac{1}{2},{\frac{1}{2}}^{-}$ and nnscc, $IJ^{P}=0,{\frac{1}{2}}^{-}$ pentaquark experimental masses vs. T-MIT calculated masses and their corresponding q values.

**Figure 5 entropy-27-00681-f005:**
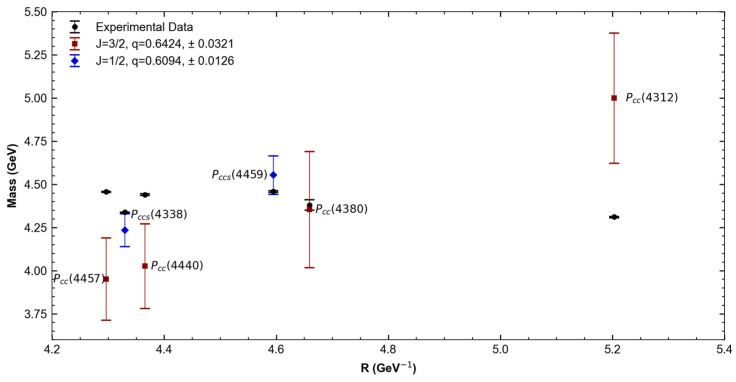
Comparison of nnncc, $IJ^{P}=\frac{1}{2},{\frac{3}{2}}^{-}$ and nnscc, $IJ^{P}=0,{\frac{1}{2}}^{-}$ pentaquark experimental masses vs. T-MIT calculated masses and their corresponding q values.

**Figure 6 entropy-27-00681-f006:**
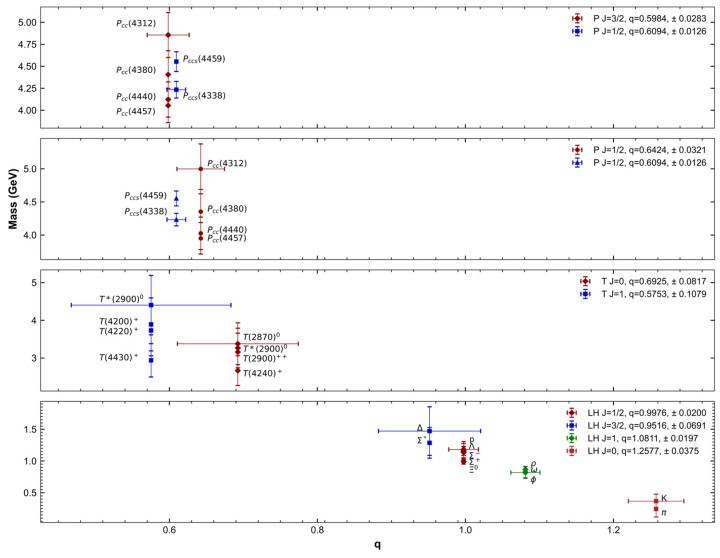
Masses of various hadrons plotted as a function of the Tsallis parameter q. The top two panels correspond to pentaquark states, the third panel to tetraquark candidates, and the bottom panel to light hadrons. Each point includes the corresponding J-value and the extracted q-parameter, along with its uncertainty.

**Table 1 entropy-27-00681-t001:** Experimental masses of light hadrons and their charge radii.

Particle	M_exp_ [GeV]	R_exp_ [fm]	R_exp_ [GeV^−1^]
p	0.938 ± 2.9 × 10^−10^	0.841 ± 0.00040	4.2621 ± 0.00202
Λ	1.116 ± 6 × 10^−6^	0.2645 ± 0.0999	1.3408 ± 0.5068
Σ^−^	1.197 ± 2.90 × 10^−5^	0.78 ± 0.34596	3.95304 ± 1.755
Σ^+^	1.189 ± 7 × 10^−5^	0.7215 ± 0.1413	3.6565 ± 0.7166
Ξ^0^	1.315 ± 0.0002	0.7071 ± 0.1413	3.5835 ± 0.7166
Δ	1.232 ± 0.002	0.95 ± 0.0199	4.8146 ± 0.1013
Ω^−^	1.67245 ± 0.00029	0.6799 ± 0.0199	3.446 ± 0.1013
ρ	0.775 ± 0.00023	0.7483 ± 0.1999	3.7923 ± 1.0136
ω	0.78266 ± 0.00013	0.7211 ± 0.2236	3.654 ± 1.1332
ϕ	1.01946 ± 0.000016	0.4582 ± 0.1414	2.3224 ± 0.7167
K	0.4936 ± 15 × 10^−5^	0.56 ± 0.03102	2.8380 ± 0.1571
π	0.139 ± 1.8 × 10^−7^	0.659 ± 0.004	3.3398 ± 0.0202

**Table 2 entropy-27-00681-t002:** Comparison of light hadron experimental masses vs. T-MIT calculated masses and their corresponding q Tsallis parameters.

Particle	M_exp_ [GeV]	M_T-MIT_ [GeV]	q
p	0.938 ± 2.9 × 10^−10^	1.1814 ± 0.0931	0.9976 ± 0.0200
Λ	1.116 ± 6 × 10^−6^	0.9890 ± 0.0376	0.9976 ± 0.0200
Σ^−^	1.197 ± 2.90 × 10^−5^	1.1605 ± 0.1459	0.9976 ± 0.0200
Σ^+^	1.189 ± 7 × 10^−5^	1.1402 ± 0.0903	0.9976 ± 0.0200
Ξ^0^	1.315 ± 0.0002	1.1352 ± 0.0887	0.9976 ± 0.0200
Δ	1.232 ± 0.002	1. 4716 ± 0. 382	0.9516 ± 0.0690
Σ*	1.67245 ± 0.00029	1. 2865 ± 0.241	0.9516 ± 0.0690
ρ	0.775 ± 0.00023	0.8185 ± 0.091	1.0811 ± 0.0196
ω	0.78266 ± 0.00013	0.8249± 0.089	1.0811 ± 0.0196
ϕ	1.01946 ± 0.000016	0.8663 ± 0.045	1.0811 ± 0.0196
K	0.4936 ± 15 × 10^−5^	0.3696 ± 0.1075	1.2577 ± 0.0375
π	0.139 ± 1.8 × 10^−7^	0.2424 ± 0.1246	1.2577 ± 0.0375

**Table 3 entropy-27-00681-t003:** Tetraquark experimental masses and bag radii.

Particle	M_exp_ [GeV]	R_MIT_ [GeV^−1^]
T*cs0(2870)^0^	2.866 ± 0.00728	4.43173 ± 0.04157
T*cs1(2900)^0^	2.904 ± 0.005099	4.6341 ± 0.0370215
T*c$\bar{s}$0(2900)^0^	2.892 ± 0.020518	4.2939 ± 0.01012
T*c$\bar{s}$0(2900)^0^	2.921 ± 0.017117	4.16028 ± 0.00739
Tc$\bar{c}$1(4200)^+^	4.196 ± 0.035355	4.17521 ± 0.212801
Tccs1(4220)^+^	4.216 ± 0.049244	4.02224 ± 0.232096
Tccs0(4240)^+^	4.239 ± 0.04846	3.49836 ± 0.144055
Tc$\bar{c}$1(4430)^+^	4.478 ± 0.015	3.19873 ± 0.0336417

**Table 4 entropy-27-00681-t004:** Tetraquark experimental masses vs. T-MIT calculated masses and their corresponding q values.

Particle	M_exp_ [GeV]	M_T-MIT_ [GeV]	q
T*cs0(2870)^0^	2.866 ± 0.00728	3.3784 ± 0.5551	0.6925 ± 0.0816
T*cs1(2900)^0^	2.904 ± 0.005099	4.4043 ± 0.7898	0.5753 ± 0.1079
T*c$\bar{s}$0(2900)^0^	2.892 ± 0.020518	3.2660 ± 0.5262	0.6925 ± 0.0816
T*c$\bar{s}$0(2900)^++^	2.921 ± 0.017117	3.1593 ± 0.4998	0.6925 ± 0.0816
Tc$\bar{c}$1(4200)^+^	4.196 ± 0.035355	3.8927 ± 0.7024	0.5753 ± 0.1079
Tccs1(4220)^+^	4.216 ± 0.049244	3.7311 ± 0.6708	0.5753 ± 0.1079
Tccs0(4240)^+^	4.239 ± 0.04846	2.6651 ± 0.3943	0.6925 ± 0.0816
Tc$\bar{c}$1(4430)^+^	4.478 ± 0.015	2.9388 ± 0.4417	0.5753 ± 0.1079

**Table 5 entropy-27-00681-t005:** Experimental masses and bag radii of nnncc, $IJ^{P}=\frac{1}{2},{\frac{1}{2}}^{-}$ and nnscc, $IJ^{P}=0,{\frac{1}{2}}^{-}$ pentaquark.

Particle	M_exp_ [GeV]	R_MIT_ [GeV^−1^]
Pcc(4312)	$4.3119_{-0.0009}^{+0.0068}$	4.75925 ± 0.04428
Pccs(4338)	4.3382 ± 0.0008	4.33 ± 0.0031
Pcc(4380)	4.38 ± 0.0300	4.40288 ± 0.13152
Pcc(4440)	$4.4403_{-0.0048}^{+0.0043}$	4.16758 ± 0.0228
Pcc(4457)	$4.4573_{-0.0018}^{+0.0041}$	4.1095 ± 0.0150
Pccs(4459)	$4.4588_{-0.0031}^{+0.0055}$	4.5947 ± 0.0323

**Table 6 entropy-27-00681-t006:** Comparison of nnncc, $IJ^{P}=\frac{1}{2},{\frac{1}{2}}^{-}$ and nnscc, $IJ^{P}=0,{\frac{1}{2}}^{-}$ pentaquark experimental masses vs. T-MIT calculated masses and their corresponding q values.

Particle	M_exp_ [GeV]	M_T-MIT_ [GeV]	q
Pcc(4312)	$4.3119_{-0.0009}^{+0.0068}$	4.8560 ± 0.2562	0.5983 ± 0.0283
Pccs(4338)	4.3382 ± 0.0008	4.2337 ± 0.0944	0.6094 ± 0.0126
Pcc(4380)	4.38 ± 0.0300	4.4053 ± 0.271	0.5983 ± 0.0283
Pcc(4440)	$4.4403_{-0.0048}^{+0.0043}$	4.1221 ± 0.2003	0.5983 ± 0.0283
Pcc(4457)	$4.4573_{-0.0018}^{+0.0041}$	4.0540 ± 0.1946	0.5983 ± 0.0283
Pccs(4459)	$4.4588_{-0.0031}^{+0.0055}$	4.5531 ± 0.1118	0.6094 ± 0.0126

**Table 7 entropy-27-00681-t007:** Experimental masses and bag radii of nnncc, $IJ^{P}=\frac{1}{2},{\frac{31}{2}}^{-}$ and nnscc, $IJ^{P}=0,{\frac{1}{2}}^{-}$ pentaquark.

Particle	M_exp_ [GeV]	R_MIT_ [GeV^−1^]
Pcc(4312)	$4.3119_{-0.0009}^{+0.0068}$	5.20317 ± 0.144
Pccs(4338)	4.3382 ± 0.0008	4.33 ± 0.0031
Pcc(4380)	4.38 ± 0.0300	4.65956 ± 0.1720
Pcc(4440)	$4.4403_{-0.0048}^{+0.0043}$	4.36568 ± 0.0274
Pcc(4457)	$4.4573_{-0.0018}^{+0.0041}$	4.29634 ± 0.0178
Pccs(4459)	$4.4588_{-0.0031}^{+0.0055}$	4.5947 ± 0.0323

**Table 8 entropy-27-00681-t008:** Experimental masses and bag radii of nnncc, $IJ^{P}=\frac{1}{2},{\frac{3}{2}}^{-}$ and nnscc, $IJ^{P}=0,{\frac{1}{2}}^{-}$ pentaquark.

Particle	M_exp_ [GeV]	M_T-MIT_ [GeV]	q
Pcc(4312)	$4.3119_{-0.0009}^{+0.0068}$	4.9992 ± 0.3771	0.6423 ± 0.0321
Pccs(4338)	4.3382 ± 0.0008	4.2337 ± 0.0944	0.6094 ± 0.0126
Pcc(4380)	4.38 ± 0.0300	4.3537 ± 0.3361	0.6423 ± 0.0321
Pcc(4440)	$4.4403_{-0.0048}^{+0.0043}$	4.0261 ± 0.2456	0.6423 ± 0.0321
Pcc(4457)	$4.4573_{-0.0018}^{+0.0041}$	3.9510 ± 0.2379	0.6423 ± 0.0321
Pccs(4459)	$4.4588_{-0.0031}^{+0.0055}$	4.5531 ± 0.1118	0.6094 ± 0.0126

## Data Availability

This manuscript has no associated data; hence, data will not be deposited. All data generated during the study are contained in this published article.

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
