# Peer review of "Masses of Hadrons, Tetraquarks, and Pentaquarks Through a Tsallis-Entropy Approach in the MIT Bag Model"

_entropy, 2025, doi:10.3390/e27070681_

Round 1
Reviewer 1 Report
Comments and Suggestions for Authors
The authors adopt a non-extensive statistics description of quark and gluons, and investigated the mass spectrum of hadrons including exotic multiquark states. As a result of the analysis, it is concluded that a non-extensive statistics can extend the MIT bag model in a meaningful way. The best-fit entropic index q depends on hadron groups, and the smaller values of q found in exotic hadron channels would imply unique quark-gluon dynamics in multiquark systems. This framework is at present far from quantitative mass predictions, but it is considered to be an interesting attempt that casts light on the non-extensive features in hadrons.
I have a few questions.
(1)
In the analysis, the charge radius is employed to fit light hadron masses, and the MIT-radius is used to reproduce exotic hadron masses. What if one uses only MIT-radii throughout the analysis? I think it's better to adopt the same procedures for light and exotic hadrons. At least, the authors should try it.
(2)
The best-fit parameters q are smaller for exotic hadrons than for light hadrons, which is one of the main conclusions and interesting. I am concerned about the q-dependence of "hadron masses" extracted using Eq.(17). Can you show some "hadron masses" plotted as a function of q fixing other adjustable parameters?
Author Response
Manuscript ID: entropy-3696280
Title: Masses of hadrons, tetraquarks, and pentaquarks through a Tsallis-entropy approach in the MIT bag model
Authors: Vanesa Fernández, Gerardo Herrera Corral, Manuel A. Matías Astorga, Jesús M. Sáenz
General Statement
We would like to thank the reviewers for their careful reading of our manuscript and their insightful and constructive comments. We have carefully revised the manuscript accordingly and provided a detailed response to each point raised below.
All changes have been incorporated into the revised version of the manuscript, with modifications highlighted as requested.
Response to Reviewer 1
Comment 1:
In the analysis, the charge radius is employed to fit light hadron masses, and the MIT-radius is used to reproduce exotic hadron masses. What if one uses only MIT-radii throughout the analysis? I think it's better to adopt the same procedures for light and exotic hadrons. At least, the authors should try it.
Response1:
It is indeed an option that provides a consistent framework for the entire analysis, placing all hadrons estimates on equal footing. However, we aim to use experimental data when available, which allows us to test our ideas against observations and real values before proceeding to more speculative yet informed values.
As a compromise, we have now added the following lines:
We use experimentally extracted radii for the light hadron masses to test our ideas against observed values. When considering radii derived from the MIT model for light hadrons as well, we found no significant changes in the overall description.
|
Particle |
MR-MIT[GeV] |
MR-Exp[GeV] |
||
|
p |
1.1766 ± 0.1030 |
1.0046 ± 0.01881 |
1.1814 ± 0.0931 |
0.9976 ± 0.0200 |
|
Λ |
1.1269 ± 0.0824 |
1.0046 ± 0.01881 |
0.9890 ± 0.0376 |
0.9976 ± 0.0200 |
|
Σ⁻ |
1.1043 ± 0.1246 |
1.0046 ± 0.01881 |
1.1605 ± 0.1459 |
0.9976 ± 0.0200 |
|
Σ⁺ |
1.1021 ± 0.0789 |
1.0046 ± 0.01881 |
1.1402 ± 0.0903 |
0.9976 ± 0.0200 |
|
Ξ⁰ |
1.0985 ± 0.0777 |
1.0046 ± 0.01881 |
1.1352 ± 0.0887 |
0.9976 ± 0.0200 |
|
Δ |
1.4678 ± 0.3211 |
0.9616 ± 0.05123 |
1. 4716 ± 0. 382 |
0.9516 ± 0.0690 |
|
Σ* |
1.3476 ± 0.2378 |
0.9616 ± 0.05123 |
1. 2865 ± 0.241 |
0.9516 ± 0.0690 |
|
0.8359 ± 0.0951 |
1.0674 ± 0.0147 |
0.8185 ± 0.091 |
1.0811 ± 0.0196 |
|
|
ω |
0.8482 ± 0.0892 |
1.0674 ± 0.0147 |
0.8249± 0.089 |
1.0811 ± 0.0196 |
|
ϕ |
0.8661 ± 0.0674 |
1.0674 ± 0.0147 |
0.8663 ± 0.045 |
1.0811 ± 0.0196 |
|
K |
0.5014 ± 0.0342 |
1.2880 ± 0.00224 |
0.3696 ± 0.1075 |
1.2577 ± 0.0375 |
|
π |
0.1371 ± 0.0097 |
1.2880 ± 0.00224 |
0.2424 ± 0.1246 |
1.2577 ± 0.0375 |
The previous table presents the values of both the masses and the q-Tsallis parameter for two cases: when radii are estimated using the MIT bag model (center column) and when experimental radii are used (right column). As shown, the results are in good agreement with the uncertainties.
We also present the plot generated using the values in the table and the corresponding radii for each case. It is important to note that in some instances, the radii differ, causing the points to shift to the left or to the right. However, as the table shows, the masses remain similar. For this reason, the plot may not be particularly illustrative.
Obtained using radii extracted from the MIT bag model
Obtained using radii derived from experimental values
All in all, we would prefer to retain the radius values derived from experimental data. This choice reflects our inclination to incorporate real measurements into our analysis. We hope the reviewer will grant us this preference.
Comment 2:
The best-fit parameters q are smaller for exotic hadrons than for light hadrons, which is one of the main conclusions and interesting. I am concerned about the q-dependence of "hadron masses" extracted using Eq.(17). Can you show some "hadron masses" plotted as a function of q fixing other adjustable parameters?
Response 2:
Thank you for the suggestion, which we have incorporated into the revised manuscript. A new figure (Figure 6) has been added to illustrate the hadron masses as a function of the q-Tsallis parameter.
We have also integrated the corresponding explanatory text:
Figure 6 shows hadron masses as a function of the q-Tsallis parameter. An overall trend is observed: lighter hadrons tend to correspond to higher q values. It is also evident that exotic states generally require lower q values for their description, suggesting that quark-gluon correlations differ significantly between mesons and baryons compared to multiquark states.
Figure 6. Masses of various hadrons plotted as a function of the Tsallis parameter q. The top two panels correspond to pentaquark states, the third panel to tetraquark candidates, and the bottom panel to light hadrons. Each point includes the corresponding J-value and the extracted q-parameter, along with its uncertainty.

Reviewer 2 Report
Comments and Suggestions for Authors
The report is attached.

Author Response
Manuscript ID: entropy-3696280
Title: Masses of hadrons, tetraquarks, and pentaquarks through a Tsallis-entropy approach in the MIT bag model
Authors: Vanesa Fernández, Gerardo Herrera Corral, Manuel A. Matías Astorga, Jesús M. Sáenz
General Statement
We would like to thank the reviewers for their careful reading of our manuscript and their insightful and constructive comments. We have carefully revised the manuscript accordingly and provided a detailed response to each point raised below.
All changes have been incorporated into the revised version of the manuscript, with modifications highlighted as requested.
Response to Reviewer 2
Comment 1:
Please give references in line 29 (masses, decay width, and etc and line 30 (recent discovery of exotic hadronic states) !
Response 1:
We thank the reviewer for this suggestion. We have expanded the references accordingly, now [1-14]:
- For line 29, discussing theoretical studies on masses, decay widths, and stability of tetraquarks and pentaquarks, we have included the recent comprehensive review by Liu et al. ( arXiv:2404.06399 [hep-ph] to appear in Reports ) , DOI:https://doi.org/10.48550/arXiv.2404.06399).
- For line 30, regarding recent experimental discoveries of exotic hadronic states, we have incorporated a broad set of recent measurements from the LHCb and CMS collaborations. These are the reports of 23 new states that have been recently observed.
These additions to the manuscript strengthen the introduction and reflect both the theoretical and experimental developments in multiquark spectroscopy.
Comment 2:
Please change [1]-> Ref.[1] in line 34 and hereafter please change the citation way!
Response 2:
We thank the reviewer for the suggestion. However, we have reviewed the MDPI Entropy author guidelines and template, which specify reference citations should be given simply as numerals in square brackets ([1], [2,3], etc.). We have, therefore, retained the MDPI formatting for consistency.
Comment 3:
Line 46 and 47: “numerous other studies follow similar approaches” — please give more appropriate references!
Response 3:
We give reference [20] for that remark, and now we have added (in the same item) the following:
Zu-Hang Lin, Chun-Sheng An, and Cheng-Rong Deng, P-wave states from diquarks
Phys Rev. D 109 056005
Which presents a similar and recent (2024) analysis over the same line. Also:
Halil Mutuk, Doubly-charged states in the dynamical diquark model,
Phys Rev. D 110 (2024) 034025
in which D mesons are used to calibrate the masses.
Finally, we add one reference for the baryon case:
Ye Xing and Ruilin Zhu, Week decays of stable double heavy tetraquark states, Phys. Rev. D98 (2018) 053005
Comment 4:
Line 50: please correct (CMI) Refs. [2,3]!
Response 4:
The reference to the Chromomagnetic Interaction Model has been corrected and now reads (with updated reference numbers): …Chromomagnetic Interaction Model (CMI) [16,17]…
Comment 5:
Line 52 please give appropriate references, Also I do not know what is related the hadronic matter in Lattice QCD with the previous paragraph because As long as I know the Lattice QCD that mentioned calculated the phase diagram of QCD. That’s why the problem comes up! So it seems this paragraph is not really connected to the hadron properties. Please clarify!
Response 5:
We included reference [25] to the statement: “Hadronic matter is also studied with Lattice QCD”
which is Section Lattice Quantum Chromodynamics of the Particle Data Group:
Review of Particle Physics
Particle Data Group, S. Navas et al. Phys. Rev. D110 (2024) 030001
We have clarified the description of the limitations of Lattice QCD related to the sign problem. This limitation highlights the need for effective models that can describe hadron properties across the entire QCD phase space. This is the one connection.
The general motivation for using effective models of hadrons, such as the MIT bag model and the one proposed here, is that they enable us to track the evolution of strongly interacting systems as a function of temperature and chemical potential. This is the second connection.
We already made this point in the paragraph next to the one under discussion, which we reproduce here for clarity:
Such modified bag models enable the study of quark matter at high temperatures and/or baryonic densities. In this regime, hadrons undergo deconfinement, giving rise to a new phase of matter known as the quark-gluon plasma [31, 32], which is of particular interest in heavy-ion collisions.
Comment 6:
Please give references for the MIT bag model in line 57! Is that correct to mentioned that the hadrons are composed by quarks and gluons, not only valence quarks (common MIT bag model) Please clarify!
Response 6:
We clarified that in our approach, quarks are treated as constituent particles confined in the bag, while gluons contribute collectively to the system’s thermodynamical quantities through the Tsallis statistical framework. The MIT bag model did consider the presence of fermions and bosons in the bag. In that sense, quarks and gluons are considered.
We have now added references [26-29], for example, reference [29]:
C.E. De Tar and J.F. Donoghue, Bag models of hadrons, Ann. Rev. Nucl. and Part. Sci. 1983 33:235
Originally, references were omitted at this point because we delve into the MIT bag model in the next section. Now, we attend to the justified request by including a classical review reference here.
Comment 7:
Line 75, is that the correct reference for the first of Bag model descriptions? As long as I know Chodos etc from MIT that suggest the model first, that’s why it is called MIT bag model, Could you please clarify!
Response7:
We respectfully disagree. As noted in this report, the original bag model was proposed by Bogoliubov. Chodos et al. later introduced the concept of bag pressure, addressing certain important aspects of Bogoliubov´s proposal and developing what became known as the MIT bag model.
The mistakenly omitted references (now [26-28]) have now appropriately included:
Chodos, R. L. Jaffe, K. Johnson, C. B. Thorn, and V. F. Weisskopf, Phys. Rev. D 9, 3471 (1974).
Chodos, R. L. Jaffe, K. Johnson, and C. B.Thorn, Phys. Rev. D10 (1974) 2599
Chodos and C. B. Thorn, Phys. Lett. 53B, 359 (1974).
It is worth noting that these authors included a remark in a note at the end of their article: Phys. Rev. D10 (1974) 2599, which we reproduce here:
“After the completion of this paper, we learned of the work of P. N. Bogoliubov [Ann. Inst. Henri Poincaré 8, 163 (1967)]. Bogoliubov considered massless quarks confined to an infinite, spherical "square well" potential of radius Ro. No confining "pressure" was introduced; instead, the "square well" was motivated as an approximation to a "self-consistent" interquark potential. R 0 was fixed by equating the total energy of the quarks to the mass of the hadron. Bogoliubov's calculations are quite similar to ours; the primary differences stem from the absence in his model of a term -g_munuB in the stress tensor, which is essential to achieve confinement in a Lorentz-covariant way. We wish to thank Dr. S.-H. H. Tye for bringing Bogoliubov's work to our attention.”
Comment 8:
In Eq.(21) what is the Z_0 variable and how did you determine the values of Z_0?
Response 8:
A feature of the MIT bag model is the introduction of the additional correction bag energy:
. It has been understood as a Casimir energy associated with the fluctuations in the empty bag. There are attempts to calculate the value of this term from first principles (e.g., J. Phys. A 31:1743-1759, 1998, and others); however, it is most often taken as a parameter with a value of approximately 2.
We use the MIT bag model to extract the radii when not available and follow the lines of the analysis in reference [34] (of the revised version of the manuscript)
DeGrand, T.; Jaffe, R.L.; Johnson, K.; Kiskis, J. Masses and other parameters of the light hadrons. Phys. Rev. D 1975, 12, 2060–2076.
In their analysis, the authors discuss the zero-point fluctuations represented in . They express that there may be a dependence on the volume of the bag and explain that they compute the zero-point energy with a cutoff. They consider a cutoff that is large in comparison to the scales of the theory and observe that the relevant quantities are insensitive to the cutoff.
Comment 9:
The interaction energies between quarks and gluons are captured in Tsallis parameter q, where the parameter values q determined by reproducing the hadron mass in Table 1 and results of q is given in Table 2. What is the physics interpretation of the q=1? Can be values of the q not equal 1? By seeing the equation (17), by taking the values of q=1, indicating the Tsallis parameter does not affect the hadron internal energy.
Response9:
The parameter q is often referred to as the entropic index or non-extensivity parameter. It quantifies the degree of non-extensivity or deviation from the standard Boltzmann Gibbs statistics.
When q=1, the Tsallis entropy reduces to the usual Boltzmann-Gibbs form, and the internal energy expression becomes equivalent to that of the standard MIT bag model. Therefore, q = 1 represents a fully extensive, non-correlated quark-gluon system, as originally assumed in the MIT model.
We state this explicitly immediately after equation (1), where we say:
From equation (1), one can obtain the BG statistics with q = 1.
When non-extensive effects arise, potentially reflecting correlations and possibly coupling among quarks and gluons not fully captured by the standard bag model. The exact interpretation of q in a reductionist approach remains unclear. It is the representation of a statistical correlation.
Notably, values of q significantly different from unity are observed in multiquark systems such as tetraquarks and pentaquarks, which may exhibit more complex internal dynamics than conventional hadrons.
We have added a clarification in Section 2.6 and expanded the discussion in the conclusions to explicitly add on the physical meaning of q, both for and .
Comment 10:
In line 191, what is the expression for B_EB? I did not see the Tsallis parameter effect on mass formula of MIT Bag, I expect it contribute through B_EB? Please clarify.
Response 10:
Please notice that we use the classical MIT bag model to extract the radii when not experimentally available, and for that, we follow the lines of the analysis in reference [34], i.e.
DeGrand, T.; Jaffe, R.L.; Johnson, K.; Kiskis, J. Masses and other parameters of the light hadrons. Phys. Rev. D 1975, 12, 2060–2076.
We use experimental data of hadron radius when available, and this allows us to test our ideas against observations and real data before proceeding to models. We then resort to the classical MIT bag model [26-28] to obtain a hadron radius that is not available from experimental measurements but does not modify the model. In other words, it should be considered a source of information that otherwise remains missing for new states of matter.
In the MIT bag model, the term represents spin interaction energy.
is the spin-binding energy that arises from the short-range chromoelectric interaction between quarks and antiquarks. This energy becomes significant only when both quarks involved are massive and move non-relativistically
As explained in the text after equation (23).
Comment 11:
In Table 2, the hadron mass predicted is variation depend on the type of hadron. Although the difference is not to much (around 10%), could you please discuss what is caused the difference? What do you think can be done to improve it??
Response 11:
We thank the reviewer for bringing this point to our attention. The small differences (~10%) between the predicted and experimental masses in Table 2 are due to simplifying assumptions, including the use of a single fitted q value per hadron group and the omission of detailed interaction terms, such as spin-spin and flavor-dependent interactions, as well as spin couplings. We now include a discussion of these limitations in Section 3.1.1. and outline possible improvements in the conclusions.
Please also consider that the measured mass values are still subject to improvement, as experimental results do change - new data analysis in the future will certainly be available -
Comment 12:
For tetraquark the values of q are rather different from the hadron in Table 2. With these values of q, what is the physical interpretation for this? Please explain!
Response 12:
We thank the reviewer for this important observation. The significantly lower q values obtained from tetraquarks (as compared to light hadrons) reflect the increasing complexity of multiquark systems. Unlike conventional hadrons, tetraquarks may involve quark-antiquark pairs, which introduce additional color, spin, and spatial correlations. These correlations can produce effective collective effects within the hadron, captured phenomenologically by the Tsallis q parameter. The lower q values suggest deviations of ideal extensive behavior, consistent with more complex internal quark-gluon dynamics in exotic states. We added a discussion of this point to Section 3.1.2.
Further studies involving a larger number of tetraquark states may provide deeper insight into the detailed dependence on quantum numbers, masses, and related properties.
Comment 13:
For pentaquark also the values of q is smaller than those for tetraquark and hadron. If the q shows the energy interaction of quark and gluons, increasing number of quarks will increase the interaction energy, meaning the values of q should be bigger. However, it happen oppositely, the values of q decreases. Please clarify! Also pentaquark and tetraquark have probability composition which are can be molecule or not. In this case, with values of q can we say something about the composition of the pentaquark or tetraquark? Is molecule or not.
Response 13:
Pentaquark masses are described using smaller q values like the case of tetraquarks. The response to the previous comment also applies here.
It is not accurate to state that q “shows the energy interaction of quarks and gluons.” The physical meaning of q is not straightforward within a reductionist framework. Rather, q is a statistical parameter that encapsulates correlations within the system. Its interpretation has been the subject of ongoing discussion. Some studies suggest that q may indicate the presence of long-range interactions, while others associate it with fractal-like structures in the system. However, these interpretations remain controversial, and there is no consensus beyond the generally accepted view that q statistically characterizes correlation among subsystems of a system.
We respectfully disagree with the reviewer´s reasoning that an “increasing number of quarks will increase the interaction energy, meaning the values of q should be bigger.” There is no established theoretical basis to support such a direct relationship.
We do agree, however, that the internal structure of tetraquarks and pentaquarks is still under investigation, particularly regarding their possible molecular nature. At this stage, based on the current results, we cannot make definitive statements on the issue.
Further studies involving a larger number of Tetraquark, Pentaquark, and Hexaquark states may shed more light on the detailed dependence of q and quantum numbers, masses, and related properties. Such an expanded analysis may also offer better avenues for exploring the internal structure of these exotic hadrons.
Comment 14:
Line 122, above Eq.(4), you cited Ref.[25], I think it is not correct reference. I know that they calculated the pressure of the proton using the energy momentum tensor. But it is not related to the gluon pressure in Eq.(4). Also there is no available in literature the pressure of proton at finite temperature. Even though there is a paper in the literature try to describe the quark matter using the proton pressure of Ref. [25]. Please cite a correct reference!
Response 14:
In the study of Burkert et al. Nature (2018), the authors extracted the quark pressure distribution within the proton as a function of radial distance from its center. This was achieved by using gravitational form factors to interpret experimental data from Deep Virtual Compton Scattering.
We agree with the reviewer that this represents a fundamentally different approach and that caution is needed when interpreting the underlying concepts. We originally cited this work as an example of efforts to understand confinement and some aspects of hadronic structure, but we now chose to withdraw the reference at this point of the exposition in light of the reviewer’s comment.
We emphasize that in this context, pressure is understood as a thermodynamic quantity derived from statistical mechanics, representing the collective behavior of quarks and gluons under confinement. This concept differs from other recent interpretations of hadronic pressure, such as the spatial pressure distribution extracted from energy-momentum tensor via Deep Virtual Compton Scattering. While their approach probes the internal mechanical structure of the proton, our model focuses on the equilibrium pressure that balances confinement in the bag. Both perspectives offer valuable yet distinct insights into hadronic pressure.
Comment 15:
What happen to Fig. 1, phase diagram if q is not equal 1.
Response 15:
We explore this case in reference [49]:
Barboza Mendoza, C.; Herrera Corral, G. Quark matter description in a Tsallis entropy approach. Eur. Phys. J. A 2019, 55, 146. https://doi.org/10.1140/epja/i2019-12834-y.
In general terms, we found that:
For the case of non-zero chemical potentials, the systems with may be associated with the weakly coupled systems while those with are more correlated. Furthermore, we find that the critical temperature for the hadron increases as the correlation between quarks and gluons increases.

Reviewer 3 Report
Comments and Suggestions for Authors
By using Tsallis non-extensive statistics to take into consideration potential non-equilibrium effects in hadronic systems, the manuscript offers a novel extension of the MIT bag model. Masses and radii of exotic states (tetraquarks and pentaquarks) and typical hadrons are computed using the model. In the work, theoretical predictions and experimental data are compared, and the Tsallis entropic parameter q is systematically fitted for various hadron types.
I have some comments:
1- Equation (3): Put the units of and R.
2-Although the model performs well for masses, it is not discussed how it applies to other features (such as decay widths and charge distributions). The paper would be strengthened by a brief discussion of these constraints.
3- the quality of figure need to improve please replace them with high quality and clear legend.
4-The work offers a strong and original addition to statistical modeling and hadron spectroscopy. The method is sound theoretically and provides insightful information. Before the manuscript is ready for publication, it would benefit from more physical interpretation, more understandable methodological description, and better presentation.
Comments on the Quality of English LanguageThough it may use some light editing to enhance grammar, flow, and professional tone, the manuscript's English is usually clear and intelligible.
Author Response
Manuscript ID: entropy-3696280
Title: Masses of hadrons, tetraquarks, and pentaquarks through a Tsallis-entropy approach in the MIT bag model
Authors: Vanesa Fernández, Gerardo Herrera Corral, Manuel A. Matías Astorga, Jesús M. Sáenz
General Statement
We would like to thank the reviewers for their careful reading of our manuscript and their insightful and constructive comments. We have carefully revised the manuscript accordingly and provided a detailed response to each point raised below.
All changes have been incorporated into the revised version of the manuscript, with modifications highlighted as requested.
Response to Reviewer 3
Comment 1:
Equation (3): Please include the units of B^{1/4} and R.
Response 1:
The bag model pressure has dimensions of energy divided by volume. In natural units where , has dimensions of mass. We use natural units throughout the derivations. In the manuscript, we express in MeV and R in femtometers (fm) and GeV-1. This clarification has been included immediately after equation (3).
Comment 2:
Although the model performs well for masses, it is not discussed how it applies to other features (such as decay widths and charge distributions). The paper would be strengthened by a brief discussion of these constraints.
Response 2
We thank the reviewer for bringing this to our attention. As suggested, we expanded the discussion in the Conclusions. In addition to noting that decay widths and charge distributions are not directly computed in our approach, we now briefly describe how extensions of the bag model, utilizing the superposition of bag states, may address decay processes, referencing [57] recent work by Geng et al. (Phys. Rev. D 102, 034033 (2020)). We have also clarified how charge contributions can be formulated within the bag framework and mentioned that this remains a possible future extension of the model presented here. Finally, we note that related applications of the Tsallis-MIT formalism have been made to study the QCD phase diagram and nucleon pressure distributions in the past.
Comment 3:
Figure quality needs improvement.
Response 3:
The resolution of all figures has been improved to enhance clarity and readability. Additionally, figures have been updated from black and white to color to distinguish data elements more clearly and enhance visual interpretation.
Comment 4:
The work offers a strong and original addition to statistical modeling and hadron spectroscopy. The method is sound theoretically and provides insightful information. Before the manuscript is ready for publication, it would benefit from more physical interpretation, more understandable methodological description, and better presentation.
Response 4:
We thank the reviewer for the positive overall evaluation and for highlighting areas for improvement.
Regarding the physical interpretation, we have added multiple clarifications in Sections 2.6, 3.1.2, 3.1.3, and 4.
Regarding the methodological description, we have expanded Section 2.6 to describe the whole procedure by which experimental data on masses and radii are used as inputs for the fitting of the Tsallis parameter q via least-squares minimization of Eq. (17). We also included a detailed explanation of how hadrons are grouped, how radii are handled, and how uncertainties are propagated. This provides a complete and reproducible account of the fitting methodology.
We have improved figure quality and added Figure 6.

Round 2
Reviewer 1 Report
Comments and Suggestions for Authors
The authors properly responded to comments and questions raised by the referees. Now I believe that the paper is worth publishing in the journal.